



**Hydroclimate variability in Scandinavia over the last millennium - insights from a**
**climate model-proxy data comparison**
Kristina Seftigen[1,2,*], Hugues Goosse[2], Francois Klein[2], Deliang Chen[1]
[1]Regional Climate Group, Department of Earth Sciences, University of Gothenburg, Gothenburg, Sweden.
[2]Georges Lemaître Centre for Earth and Climate Research (TECLIM), Earth and Life Institute, Université
catholique de Louvain (UCL), Belgium.
[*]Corresponding author:
E-mail address: kristina.seftigen@gvc.gu.se
**Abstract**
The integration of climate proxy information with General Circulation Model (GCM) results
offers considerable potential for deriving greater understanding of the mechanisms underlying
climate variability, as well as unique opportunities for out-of-sample evaluations of model
performance. In this study, we combine insights from a new tree-ring hydroclimate
reconstruction from Scandinavian with projections from a suite of forced transient simulations
of the last millennium and historical intervals from the CMIP5 and PMIP3 archives. Model
simulations and proxy reconstruction data are found to broadly agree on the modes of
atmospheric variability that produces droughts/pluvials in the region. But despite these
dynamical similarities, large differences between simulated and reconstructed hydroclimate
time series remain. We find simulated interannual components of variability to be
overestimated, while the multidecadal/longer timescale components generally are too weak.
Specifically, summertime moisture variability and temperature are weakly negatively
associated at inter-annual timescales but positively correlated at decadal timescales, revealed
from observational and proxy evidences. On this background, the CMIP5/PMIP3 simulated
timescale dependent relationship between regional precipitation and temperature is
considerably biased, because the short-term negative association is overestimated, and the
long-term relationship is significantly underestimated. The lack of adequate understanding for
mechanisms linking temperature and moisture supply on longer timescales has important
implication for future projections. Weak multidecadal variability in models also implies that
inference about future persistent droughts and pluvials based on the latest generation global
climate models will likely underestimate the true risk of these events.



## 1. Introduction

Among the current key priorities in climate research is a more comprehensive understanding of changes in regional- to continental-scale hydroclimate in response to rising levels of atmospheric greenhouse gases on time scales ranging from decades to centuries (Wu et al., 2013; Hegerl et al., 2015). Delineating the role of internal variability and natural forcing, and its contribution to the anthropogenically forced twentieth century climate (Zhang et al., 2007; Sarojini et al., 2016), is immensely important for attributing past and predicting future trajectories in the hydrological cycle, and for strategic approaches to adaptation and planning. Sparse observational evidences limits possibilities of providing tight constraints on the long-term behavior of the climate system. The longest instrumental records (~150-200 years) are too short to fully sample modes of variability that are either rare or occur on multidecadal-to-centennial timescales. This motivates the development of paleoclimatic proxy reconstructions, which extends the observational baseline into the longer spectrum of climate variability and provides a framework to consider both internal and forced climate changes.

Considerable advancements have recently been made in developing tree-ring estimates of late Holocene hydroclimate variability across Scandinavia (Seftigen et al., 2014; Cook et al., 2015). Being located in the high-latitude boreal zone, Scandinavia is well suited for dendroclimatological studies and has a long tradition of climate and environmental research using tree-ring data (Linderholm et al., 2010). The use of tree-ring proxy evidence to study natural hydroclimate variability has however long been secondary when compared to the scientific attention focused on providing local/regional reconstructions (Gunnarson et al., 2011; Esper et al., 2012; McCarroll et al., 2013; Linderholm et al., 2014) and methodologies (Björklund et al., 2012; 2014) to study temperature variability over the last several millennia. Much of the tree-ring research at moisture-limited sites have until recently been limited to a handful of exploratory papers (Helama and Lindholm, 2003; Linderholm et al., 2004; Jönsson and Nilsson, 2009; Drobyshev et al., 2011; Seftigen et al., 2013) that generally develop one or few chronologies to provide local precipitation/drought histories. These studies, together with a steadily growing collection of high-latitude moisture sensitive tree-ring records (e.g., Seftigen et al., 2015), now serves as a basis for new possibilities to expand the detail and accuracy with which the history of Northern European moisture conditions can be described. A recent milestone in the field include the development of the "Old World Drought Atlas" ("OWDA", Cook et al., 2015), a set of tree-ring reconstructed year-to-year maps that provide temporal *and* spatial details of droughts and wetness in the last millennium across Europe,



including Scandinavia. The OWDA has been used to elucidate hydroclimatic blueprints of the
Medieval Climate Anomaly (MCA, ~1000-1200 CE). Aligning with prior findings (Helama et
al., 2009), the atlas reveals the occurrence of so-called megadroughts in large portions of
continental north-central Europe and southern Scandinavia during the MCA period.
Interestingly, MCA and other "Old World" droughts seem to coincide with the timing of
some severe and persistent droughts documented in the climate history of North America.
While this suggests the presence of some common driving mechanisms across the North
Atlantic, being possibly related to variations in the Atlantic Ocean SST or/and the North
Atlantic Oscillation (Feng et al., 2011; Oglesby et al., 2012), the cause of these megadroughts
remains to be an open question.
While the proxy reconstructions undoubtedly play a pivotal role in unraveling
statistical qualities of past climate, they are, alone, not able to provide a comprehensive view
of the underlying physics governing the climate system. The forced-transient simulations over
the last millennium from fully coupled general circulation models (GCMs) (Taylor et al.,
2012) therefore offer an important complementary approach to the empirical analyses of
proxy estimates. Paleoclimate reconstructions provide an observational basis that spans
beyond current climate conditions that were used in developing and tuning such numerical
models, thus allowing for out-of-sample evaluations of the models' predictive power. The
models, on the other hand, can be used to explore the dynamics that have driven climate
variability in the past.
This paper builds on previous tree-ring analyses (Seftigen et al., 2014; 2015) and
aims at employing a paleoclimate-data model comparison framework to further explore the
drivers and dynamics of drought/pluvials across Northern Europe. We analyze an ensemble of
six state-of-the-art GCMs from the Past Model Intercomparison Phase 3 (Schmidt et al., 2011
- PMIP3) and the Coupled Model Intercomparison Phase 5 (Taylor et al., 2012 - CMIP5) and
compare them to a new regional tree-ring-based proxy reconstruction of drought and wetness,
spanning the last millennium of the Common Era (CE). A combined data approach is used to
(1) evaluate to what extent the GCMs are capable in reproducing the key features of the
paleoclimate record, and (2) to estimate the role of external forcing versus internal variability
in driving the hydroclimatic changes regionally. Inter-annual and decadal/longer-term
relationships between hydroclimate, and the two key components of rainfall and surface
temperature, are also briefly explored and the ability of the CMIP5/PMIP3 models to simulate
the mechanisms by which the regional hydroclimate is constrained by these two variables are
evaluated.  The collective proxy-model data assessment will help to increase our





understanding of decadal/longer climate dynamics in regions and to evaluate the ability of the
state-of-the-art GCMs to simulate realistic future hydroclimatology regionally and across a
variety of different timescales.
The paper is structured as follows. Sect. 2 reviews the methods and describes the
paleoclimate and CMIP5/PMIP3 datasets. Subsequent analyses concentrates on comparing the
GCM simulations with the proxy based hydroclimate reconstructions (sect. 3), and delineating
the role of external (sect. 4) and internal (sect. 5) sources of variability over the last
millennium. The principal results and the implication of this study are discussed in sect. 6.
**2. Data and methods**
**2.1 CMIP5/PMIP3 simulations**
Simulations with six models (CESM1, CCSM4, IPSL-CM5A-LR, MIROC-ESM, MPI-ESM-
P, BCC-CSM1-1) contributing to the Coupled and Paleo Model Intercomparison Projects
Phases (CMIP3/PMIP3) (Schmidt et al., 2011; Taylor et al., 2012) have been used (Table I).
The analyses were restricted to models that have available complete monthly precipitation and
temperature variables spanning the last millennium (850-1849 CE) through historical (1850-
2005 CE) time intervals. The six millennium simulations were forced with reconstructed
solar, volcanic, greenhouse gas (GHG) and aerosol forcing, and partly land use changes,
whereas the historical simulations included natural and anthropogenic forcing (Schmidt et al.,
2011; Taylor et al., 2012). Except for CESM1, the analyses were limited to the first r1i1p1
ensemble member. Supplementary information (sect. S1, Fig. S1) provide an evaluation of six
selected model rainfall and temperature simulations against instrumental reference data
focusing on the northern European sector.
**2.2. Proxy data**
Building on an existing compilation that has previously been used to derive regional
spatiotemporal drought climatology (Seftigen et al., 2014; 2015), we analyzed a network of
27 *Pinus sylvestris* L. tree-ring width (TRW) chronologies from southern Scandinavia (Fig.
1). The start dates of the chronologies varied across the collection, ranging from 532 to 1790
CE (Table II). All chronologies extended at least to year 1995. In order to reduce the risk of
natural/anthropogenic disturbance signal from inflicting non-climate noise upon the
reconstruction, the tree-ring data has been standardized in previous research (Seftigen et al.,
2014) by using a flexible "data-adaptive" method of standardization (Cook et al., 1995). This



has limited the degree to which longer-timescale climate information can be extracted.
Therefore, rather than using the already available hydroclimate reconstruction provided in
Seftigen et al. (2014), we have here re-processed the TRW collection with the newest signal-
free (SF) method of standardization (Melvin and Briffa, 2008), which has the capacity of
preserving long-term variability due to climate changes. The standardization was performed
with the ARSTAN software (Cook and Krusic, 2005). Chronologies combining living and
historical/subfossil material were standardized with a regional curve standardization (RCS)
approach (Briffa et al., 1992), applying a single RCS curve without any pith-offset
adjustments to detrend all series. To avoid spurious growth trends in the resulting RCS
chronologies stemming from a modern sample bias (Briffa and Melvin, 2011), tree-ring
datasets based only on living trees were standardized using the SF method in combination
with an age-dependent smoothing spline applied individually to each series. Prior to the
standardization, the modern chronology data were high-pass filtered and subsequently
grouped by means of a S-mode principal component analysis over the common interval (1792
– 1996 CE). The resulting eigenvector loadings are provided in supplemental material (Fig.
S2) and describe the major modes of high-frequency variability within the multiple modern
chronologies composing the dataset. The subdivision of the chronologies essentially identified
an east-west pattern, broadly corresponding to sub-regional differences in topography and
climate across the study domain. This suggested that the sub-regional tree-growth coherence
at high frequencies was driven by climate. Hence, it would be rational to expect a common,
climatically induced, growth variability also at the medium-frequency time scales, while any
disparities in the sub-regional tree-growth signal are likely mostly non-climatic in origin (i.e.
local site management practices, stand dynamics or other 'random' site-specific disturbances).
Therefore, in order to remove or minimize undesirable non-climatic noise upon our dataset,
modern tree-ring series were first merged group-wise as identified by the first four principal
components and subsequently detrended as four separate 'batches' using the SF method. The
standard version of the resulting tree-growth indices were subsequently separated and
averaged for each site to produce individual site chronologies. This procedure enabled us to
retain any shared, sub-regional, growth-forcing signal while removing site-specific medium-
to high-frequency noise.

Final data were adjusted to reduce the variance bias stemming from varying sample

size trough time (Frank et al., 2006). The resulting chronologies were truncated where the
Expressed Population Signal (EPS) (Wigley et al., 1984) dropped below the 0.85 threshold,
or, in case of the longer chronologies, at year 1000 CE. The median segment length (MSL) of



all the chronologies (Table II) ranged between 74 and 357 years, and the median MSL across
all sites was 197 years. Although a precise quantification of returned frequency variance in
the final SF detrended tree-ring chronologies was not straightforward, the median MSL
suggested that it should be possible to use the network to reconstruct climate variability at
time scales up to ~200 years.
**2.3. Regional hydroclimatology**
The CMIP5/PMIP3 inter-model spread in spatial resolution and sophistication of soil
moisture schemes makes meaningful inter-model comparison difficult. To bypass some of
these challenges, the Standardized Precipitation Evapotranspiration Index (SPEI) (Vicente-
Serrano et al., 2013) was used to characterize the regional hydroclimatology across the study
domain. The SPEI, a commonly used metric of soil moisture balance, has successfully been
used as a target variable in several prior tree-ring reconstructions (e.g., Seftigen et al., 2014;
2015). The index is not a state variable but rather an offline metric of the surface moisture
balance that can be consistently derived across models and therefore provide standard
measure of hydroclimatic variability across GCMs. The computation of the index is based on
normalized monthly climatic water balance, i.e. cumulative precipitation minus potential
evapotranspiration (PET), summed over multiple time scales and computed as standard
deviations with respect to long-term mean (Vicente-Serrano et al., 2010). The PET was here
estimated with the Thornthwaite approach (Thornthwaite, 1948). The method requires surface
temperature and latitude data only, and has therefore frequently been used for PET
computations over the historical period. Moreover, the choice of methods is motivated by the
larger confidence that is placed on GCM simulations of temperature compared to other
variables (vapor pressure, wind speed, net radiation, etc.) that are required for more physically
based parameterizations of PET. At each grid point, model SPEI were derived from estimated
PET and simulated rainfall over the past1000 and historical periods and then standardized
against the 1901-2005 normalization period using the SPEI R package version 1.6 (Vicente-
Serrano et al., 2010).

The proxy dataset was generated by a point-by-point regression (PPR) methodology

that was applied to the TRW network to produce a SPEI reconstruction spanning the past
millennium. The climate field reconstruction method is based on principal component
regression procedure using the TRW chronologies as potential predictors to develop a set of
nested multivariate stepwise regression models (see Cook et al., 1999 for details). Here we
employed the same calibration/validation scheme, predictor selection and pre-processing steps



as previously described in Seftigen et al. (2015). We performed a full period calibration over
the 1901-1995 period of TRW/climate data overlap, and a conventional split period
calibration/validation procedure (1901-1948 and 1949-1995 periods) for an independent
validation of the SPEI estimates. Each nest was centered and scaled to have the same mean
and variance as the observational data in the calibration period. The instrumental SPEI target
field for the reconstruction was computed from the CRU TS 3.22 (Harris et al., 2014) 0.5°
latitude x 0.5° longitude gridded rainfall and temperature datasets over the southern portion of
Scandinavia (55° - 65° N and 5° - 30° E) (Fig. 1), using the same conventions as described
above. Simple correlation analysis conclusively demonstrated a short-term early summer
moisture sensitivity of the TRW records over most of the study domain (Fig. S3). Based on
these findings, we selected June SPEI, aggregated over a 2-month time scale, as the target
season data for the reconstruction. A final regional time series was averaged from grid points
where the calibration regression models explained at least 20% of instrumental variance and
the reduction of error (RE) and coefficient of efficiency (CE) (National Research Council,
2006) verifications metrics exceeded the generally accepted threshold value of zero across all
nests (N = 521 grid points). The mean tree-ring hydroclimate reconstruction (henceforth
ScandH17) and the corresponding instrumental target dataset are shown in Fig. 1, and a
validation of the reconstruction against 20th century instrumental data that have been withheld
from the calibration is provided in supplementary materials (Fig. S4). Results are variable
depending on the calibration/validation period used; the validation and calibration statistics
are stronger for the 1901-1948 period and substantially weaker for the 1949-1995 period. The
most recent and well-replicated nests (mid-1600s to present) are generally explaining the
greatest amount of instrumental variance ($R^2 > 40\%$ for the majority of the grid points). A
loss of grid cells with declining proxy availability and a drop in reconstruction skill is
occurring prior to the late-1400s and subsequently in the 1200s. Point-wise correlation with
gridded instrumental SPEI dataset shows that ScandH17 is representative for a larger area in
southern and central Scandinavia with a correlation 'hot spot' exceeding 0.6 (Fig. 1).
**2.4. Analyses**
The new proxy-based reconstruction was used to assess the temporal evolution of droughts
and pluvials over the last millennium and to elucidate the mechanisms that govern
hydroclimate changes in the northern European sector ranging from interannual to
multidecadal time scales. We compared regional hydroclimate time-series with the primarily
variables governing the moisture balance: precipitation (which supplies moisture) and



temperature (which modulates potential evapotranspiration in our method) (sect. 5). The
CMIP5/PMIP3 hydroclimate was contrasted against corresponding last-millennium and
historical simulations of temperature and precipitation. As there are no independent, annually
resolved, proxy reconstructions of rainfall variability currently available for the region, we
only included temperature estimates in the comparison with ScandH17. For this purpose, the
previously published Linderholm et al. (2014) (hereafter ScandT14) summer temperature
reconstruction was used. The two reconstructions ScandH17 and ScandT14 share no common
predictors and are thus fully independent, which ensures that any circular statement in the
comparison can be ruled out. The ScandT14 record is based on tree-ring maximum density
(MXD) and blue intensity data from central-northern Scandinavia and is in terms of signal
strength and preserved multi-centennial scale variability one of the best temperature
reconstructions currently available for the region.

Furthermore, we extended our analyses to the model domain using the methodology

of paleoclimate data-model comparison. There were three main components to the combined
approach. Firstly, we evaluated the consistency in various datasets and assessed whether the
CMIP5/PMIP3 simulations have similar statistical properties as the reconstruction (sect. 3).
Spectral and spectral coherency analyses were performed in two ways. The first is the multi-
taper approach (Thomson, 1982) based on 4 tapers, where a Monte-Carlo procedure is used to
estimate phase 95% confidence limits. We also used the wavelet cohere analyses available in
the Grinsted et al. (2004) MATLAB package to assess the frequency dependent relationships
and phasing between various datasets.

Secondly, we used the Superposed Epoch Analysis (SEA) (Haurwitz and Brier,

1981) to evaluate the influence of volcanic aerosol forcing on hydroclimate, temperature and
precipitation of the Scandinavian region at inter-annual time scales (sect. 4). For the last
millennium, monthly mean volcanic forcing series were obtained from three different sources:
Gao et al. (2008), Crowley and Unterman (2013) and Sigl et al. (2015) datasets. We note that
the former two forcings have been used as the boundary conditions for the last millennium
CMIP5/PMIP3 simulations. The length of the proxy and model data allowed us to include sets
of the 20 largest eruptions since 1100 CE (Table III) from the annual forcing series to assess
the mean response. For each series and eruption, anomalies for ten post-eruption years were
computed relative to a five-year pre-eruption mean. The confidence intervals around the
composite responses were determined using a Monte Carlo block resampling (N = 10 000) of
the actual event year windows (see Adams et al., 2003 for details).

Thirdly, we evaluated the skill of the models to represent the dynamics that drive the



variability in hydroclimate of the Scandinavian region by establishing a link between
simulated and reconstructed SPEI series and fields of mean sea level pressure (MSLP) over
the Atlantic-European sector (sect. 5). Grid point correlations were computed to assess the
spatial features and the strength of the teleconnections patterns over the modern era (1950-
2005 CE). The analysis was also extended over the last millennium (1000-1849 CE) to
investigate the nature of teleconnection stability without the influence of anthropogenic
forcing. The gridded monthly instrumental HadSLP2 dataset spanning 1850-present (Allan
and Ansell, 2006) was used for comparison with observed and proxy-based estimates of
hydroclimate.
**3. Modeled and reconstructed hydroclimate series**
The regional warm season hydroclimate variability averaged across the six CMIP5/PMIP3
models together with the new ScandH17 proxy reconstruction over the last millennium are
shown in Fig. 2a-b. Individual model SPEI time series are displayed in Fig. 2c-h. All data
have been normalized and centred over the common interval from 1000 to 1995 CE, since this
first joint proxy-model comparison focuses on the common relative changes rather than on the
magnitude and the absolute values. A simple visual comparison reveals that the models and
the reconstruction have generally little agreement in the variance structure and trends. The
reconstruction is dominated by a large decadal-to-multidecadal variability while the
multimodel mean is relatively flat at these time scales. There are some common features in
some of the GCMs and the proxy datasets though (Fig. 2c-h), e.g., the drying in the 19[th]
century, but these are rare when the full millennium is considered and are likely occurring by
chance. The historical interval in the proxy record is characterized by a drought in the mid-
1800s and a gradual increase in wetness over the 20[th] century, while, with the exception of
short dry episode in the early-1900s, there is no long-term trend in the multimodel mean over
the modern era.

The very low correlation at inter-annual time scales is to be expected, as the internal

variations in the various records represent different realizations of the climate system, which
is to a very large extent chaotic at that time scale. The response of each ensemble member to a
strong external forcing applied to the model would nevertheless ideally agree (i.e. external
punctual perturbations such as volcanic eruptions could induce a coherent short-term
response, see sect. 4). Averaging across models or over multiple ensemble members will
reduce the contribution from stochastic variability so that the remaining signal can come
closer to the model response to external forcing. The comparison between ScandH17 and the



multimodel assemble mean reveal, however, no statistically significant agreement between
the series, neither on the interannual nor on decadal timescales, suggesting that the simulated
hydroclimate changes are not strongly tied to exogenous forcing. Moreover, we found no
statistically significant correlation between the different ensemble members in the same
model  (CESM1) (Fig. 2c), which is the only model providing multiple ensemble members
(the only difference among these being the air temperature at the start of each ensemble
member (Otto-Bliesner et al., 2016) over the historical and past millennium intervals). The
poor overlap between CESM1 ensemble members as well as the individual GCM simulations
over the past millennium (despite the use of largely similar forcing series to drive the
simulations) is indicative of a larger contribution from internal variability on simulated
drought/pluvial occurrence than from changes in exogenous forcing.
We compare the spectral properties of the six individual CMIP5/PMIP3 models to
the ScandH17 reconstruction, which allows for a general evaluation of potential frequency
biases. Fig. 3a confirm that the underlying spectrum of reconstructed hydroclimate variability
is significantly redder on decadal-centennial timescales than indicated by the simulated SPEI.
In contrast, more hydroclimate variance is concentrated on interannual timescales in the
CMIP5/PMIP3 archive than in ScandH17 reconstruction. At frequency bands < 8 years, the
power spectral range of most models is systematically above the confidence interval of
ScandH17. As a complementary analysis, the numbers of reconstructed and simulated multi-
year hydroclimate anomalies greater than a threshold length are compared (Fig. 3d). It is clear
that the characteristics of the paleoclimate data are generally not present in the GCM
simulations considered here, which suggests that the models are underestimating the risk of
persistent multi-year droughts and pluvials in the region. We also consider the agreement
between simulated and reconstructed (ScandT14) temperature data in terms of their spectral
properties (Fig. 3b). Although the degree of agreement is higher than for hydroclimatology
and most models lie within the reconstruction confidence bands, there are some models that
have more variance than the reconstruction at periods < 10 years.
**4. External sources of variability**
Large explosive volcanic eruptions are an important natural radiative forcing mechanism at
timescales ranging from seasons to decades (Shindell et al., 2004; Gleckler et al., 2006). The
imposed perturbation on the climate system by such events will depend on the nature of the
eruption, the magnitude of change in the energy entering the earth's atmosphere, the
background climate and internal variability, latitude and season. Analysis of observational



data (Shindell et al., 2004), tree-ring records (D'Arrigo et al., 2013) and model simulations
(Anchukaitis et al., 2010) indicate a considerable spatial variability in the dynamical response
of the climate system to volcanic forcing, with some regions experience surface and
tropospheric cooling effects and other regions showing no significant change or even
warming effect. Here, we assess the magnitude and timing of Scandinavian summer
temperature, rainfall and hydroclimate response to short-term radiative cooling due to
volcanic aerosols.

A peak cooling is observed one year after the eruption, both in ScandT14 and in the

CMIP5/PMIP3 composite average, for all the three forcings considered (Fig. 4). In addition,
there is a significant cooling in the year of the event for the Crowley and Unterman (2013)
and Sigl et al. (2015) lists. ScandT14 reveal a marginally greater cooling (2.0 °C, mean of the
three event lists) than the model average (1.8 °C) one year after the eruption. Remarkably,
there is a high degree of similarity in the proxy and in the GCMs not only in terms of the
signal timing and the magnitude of the cooling response, but also the rate of recovery. A
complete recovery after the volcanic cooling is found two years after the eruption,
independent of the forcing list. These results are generally consistent with prior studies
(Fischer et al., 2007; Jones et al., 2013; McCarroll et al., 2013) highlighting the importance of
explosive volcanism as an external driver of Northern European temperature variability. They
also provide a relevant test of the model to radiation perturbations. The agreement between
the model simulations and proxy data demonstrates the credibility of the models.

Existing research on the response of high-latitude rainfall and hydroclimate to

volcanism is limited (in part because high resolution moisture sensitive proxy records are
sparse or unavailable). Fischer et al. (2007) found a weak tendency to drying conditions over
southern/central Scandinavia in the summer of year 0 and year 1 after the eruption.
Circulation changes to the surface cooling were shown to modulate the directly forced
response. On continental and global scales, both observational and modeling studies have
found a decrease in precipitation (Iles et al., 2013) and streamflow (Iles and Hegerl, 2015) in
response to large explosive eruptions, particularly in climatologically humid regions (Carley
and Gabriele, 2014). The short-term drying is caused by a reduction in incoming solar
radiation reaching the surface, which reduces evaporation, whilst the widespread cooling
stabilized the atmosphere and lowers its water holding capacity (Bala et al., 2008). Here, we
apply SEA on ScandH17 and simulated SPEI and precipitation to examine the influence of
volcanism on Scandinavian moisture availability. A statistically significant reduction in
simulated rainfall is observed for all event lists, ranging between the year of the event



(Crowley and Unterman, 2013 dataset) and up to two years (Sigl et al., 2015 dataset)
following the eruption. We find, however, that the precipitation signal is less consistent across
the six CMIP5/PMIP3 models than the cooling effect observed in the simulated temperature
series.

The SEA on SPEI time series reveals a statistically significant drying after large

volcanism. However, the response is more muted than the response of temperature and
rainfall separately. Moreover, the agreement between proxy data and the model composite
average is weak and there are large inconsistencies between the different forcing records.
ScandH17 show a progressive transition from wet conditions in the event year and preceding
years to dryer conditions in the consecutive years with significant dry anomalies five
(Crowley and Unterman, 2013 dataset) and seven years (Sigl et al., 2015; Gao et al., 2008
datasets) after the perturbation. For the CMIP5/PMIP3 multimodel multi-eruption average,
only the fifth year after the eruption (Crowley and Unterman, 2013 list) is found to be
significantly drier than the adjacent years.

The observed weak influence of volcanic forcing on the hydroclimate of the region

can be explained by various factors. For example, our results reveal that GCM simulated post-
volcanic cooling remains significant for about two years and matches the timescale of the
post-volcanic rainfall decrease. Since the SPEI accounts for both supply and demand changes,
the net effect would be such that the temperature-driven PET decrease counter a substantial
fraction of the precipitation-driven drying, thus producing SPEI values near neutral.
Furthermore, the muted response of ScandH17 may arise from autocorrelated biological
memory in the TRW data (Esper et al., 2015). The high year-to-year persistence may bias its
ability to estimate the abruptness and severity of climatic extremes caused by volcanic
cooling. The tree-ring MXD and the blue intensity parameters have, in contrast, been
suggested to be superior TRW for recording short term climate perturbations (Wilson et al.,
2016), which is likely the reason why the response of ScandT14 is more immediate than that
of ScandH17.
**5. Internal sources of variability**
If the regional hydroclimate variability is indeed dominated by internally generated stochastic
components of variability (see sect. 3), atmospheric circulation changes can be the key
process shaping regional patterns of moisture availability. Advancing our understanding of
the range, stability and strength of teleconnection behavior (defined here as the correlation
between hydroclimate and MSLP over the Atlantic-European sector) and its coupling to





regional hydroclimate would provide an improved understanding of drought/pluvial dynamics
and associated uncertainty. In this section, we first explore major modes of atmosphere
variability that impact summertime northern European hydroclimatology. We also assess
more extensively the contribution of atmospheric processes (and possible land-atmosphere
interactions) by investigating the couplings between hydroclimate and arguably the two most
critical variables of the terrestrial climate and the hydrological cycle: precipitation and
temperature.

To determine the role of teleconnections, correlation of MSLP fields with the

hydroclimatic variables over the recent 50 years of the post-industrial era were computed.
Results are shown in Fig. 5. As expected, we find that atmospheric dynamics have a
significant role in climate variability in the region: a strong correlation with regional
hydroclimate is found when MSLP in concurrent months (i.e. May-June) is considered. The
results show that the proxy based and CMIP5/PMIP3 simulated dynamics are largely
consistent with those in the instrumental record, indicating that both the proxy and the models
contain to some degree realistic teleconnections. A consistent feature across the datasets is a
tripole structure that would favor increased moisture supply into the Scandinavian region. The
structure is characterized by anomalous cyclonic conditions across Scandinavia and high-
pressure systems extending over Iceland-Greenland and, albeit less pronounced, over
European Russia - Central Asia. Out of the six CMIP5/PMIP3 models, MIROC-ESM is the
one showing the largest discrepancy with the major spatial features of the observed
correlation map, by failing to reproduce the anti-cyclonic pattern over Iceland-Greenland.
Additionally, MIROC-ESM and also CCSM4 show a meridional and zonal shift of the
European Russia - Central Asia high-pressure structure towards the Mediterranean region.

Atmospheric circulation has been identified as key contributor to recent changes in

the climate of Europe in both summer and winter (van Oldenborgh and Van Ulden, 2003;
Jones and Lister, 2009). To assess the stationary of observed MSLP patterns, the analysis
was repeated for the pre-industrial last millennium (1000-1849 CE) period (Fig. 6). The
exercise was restricted to five GCMs for which simulated MSLP was available for the pre-
industrial era (BCC-CSM1-1 was not included). The simulated dynamical relationships were
found to be largely stable for all five models, being consistent with observed correlations
patterns in the modern era. This suggests a weak influence of anthropogenic forcing on the
structure of the dynamical drivers of Scandinavian hydroclimate. In addition to raw data,
correlation analysis with 10-year low-passed data was also completed for the pre-industrial
period with the purpose to elucidate the drivers of multidecadal hydroclimate variability. We



find similar, yet weaker, correlation patterns as compared to the high-frequency variations
(results not shown).

Precipitation and temperature are the two key variables of the hydrological cycle.

Quantifying the covariability between these two variables across various timescales, and the
mechanisms that control and modulate it, is therefore of great interest to the study of regional
processes on surface energy and water budgets. While past studies have investigated the
relationship between temperature and moisture supply in various regions on daily, seasonal
and interannual timescales (Adler et al., 2008; Berg et al., 2015; Trenberth, 2011; Madden
and Williams, 1978), the nature of concurrent multidecadal/ long-term relationship is still far
from being clear. A collective comparison of the new hydroclimate reconstruction with the
recently published Linderholm et al. (2014- ScandT14) fully independent warm-season
temperature record for Scandinavia is provided in Figs. 7 and 8, in conjunction with the
CMIP5/PMIP3 simulations of temperature and rainfall. On interannual timescales, five out of
six GCMs show a significant ($p < 0.05$) negative association between simulated interannual
temperature and rainfall, with correlation coefficients ranging between $r = -0.12$ and $-0.29$
(1000-2005 CE period). The presence of an anticorrelation on interannual timescales is also
found in the instrumental and proxy records, although the anticorrelation is significant in the
instrumental record only (Fig. S5).

Notably, a frequency dependent relationship between the ScandH17 and ScandT14

reconstructions is found. While there is a negative relationship between the two on a year-to-
year basis, a simple visual comparison of the two reconstructions shows that they are mostly
in phase on decadal and longer timescales (Fig. 7). These results are corroborated by the
cross-wavelet coherency analysis (Fig. 8a), revealing that the two reconstructions share
significant ($p < 0.05$) in phase variance in multidecadal frequency throughout most of the last
millennium. The coupling seems to arise from overlap in shared frequencies at wavelengths
longer than ~ 50 years (c.f. Fig. 3). The observed frequency-dependent shift of the
relationship thus suggests that cool summers are likely to be rainy summers on a year-to-year
basis, while over longer time, warm decades tend to be wet decades in Scandinavia. Notably,
our results reveal that the proxy reconstructions and the CMIP5/PMIP3 models portray
considerably different relationships between temperature and moisture supply in Scandinavia
on longer timescales. We find that the majority of the CMIP5/PMIP3 models are either
underestimating or even lacking the positive association between temperature and moisture
supply (Fig. 8b - h). The discrepancy appears to arise largely as the result of the spectral
inconsistencies among the model and proxy datasets (see sect. 3). While the modeled



interannual components of variability are overestimated, the decadal/longer timescale
components are generally too weak (Fig. 3).

The observed time-dependent shift of the relationship between regional temperature

and moisture availability suggests that different mechanisms governing the climate system
might be operating at high (interannual) and low (decadal/longer) frequencies, respectively.
The previously discussed strong link between inter-annual regional hydroclimate variability
and atmospheric pressure patterns suggests that atmospheric dynamics is likely a dominant
driver of hydroclimate in the northern European sector on interannual basis. The inverse
covariability between warm-season temperature and moisture supply may arise from
synoptic-scale correspondence between reduced cloud cover/rainfall and increased incoming
shortwave radiation warming the surface during clear sky conditions. In addition, soil
moisture exert a strong influence on the allocation of available energy between latent and
sensible heating, especially in the warm-season (Seneviratne et al., 2010). Reduced soil
moisture, for example, is associated with reduced latent heat flux and thus increased sensible
heating and higher air temperatures near the surface. Resulting positive feedbacks of a
modified surface heat flux partitioning on cloud cover and radiation (Gentine et al., 2013) and
large-scale circulation (Haarsma et al., 2009) could further strengthen the influence of rainfall
variability on the thermal climate.

The positive association between temperature and moisture supply that is found on

decadal-to-multidecadal timescales imply that the long-term regional hydroclimate variability
is more sensitive to changes in moisture supply (precipitation) rather than to increased
evaporative demand due to warming. It also suggests that the regional moisture balance might
be favored by the Clausius-Clapeyron relation (Allen and Ingram, 2002), prescribing an
increase in rainfall and intensity of the hydrological cycle during warmer periods in the past
millennium. This is generally referred to as 'wet-get-wetter'/'dry-get-dryer' mechanism and is
attributed to thermodynamics processes (Held and Soden, 2006). In the absence of changes in
atmospheric circulation, changes in net moisture supply with warming are related to change in
moisture content of the atmosphere. It presupposes that existing circulations will transport
more moisture into mesic regions of the globe (e.g., tropics and the mid- to latitudes of
Northern Hemisphere), whilst dry regions (e.g., subtropics) will get even dryer, with the
fractional change determined by Clausius-Clapeyron relation. In contrast to the proxy records,
the model composite average reveals a twentieth-century temperature and rainfall increase yet
little change in hydroclimate (Fig. 7b). The multimodel assessment implies that natural
variability plays only a subsidiary role in recent changes and that forcing from anthropogenic



greenhouse gases (GHG) may have played a more important role (as previously discussed, the
effect of GHG-forcing on interannual teleconnection patterns in the modern era seems to be
weak). Moreover, the absences of any significant trend in simulated SPEI series indicates that
the gains in moisture from increased precipitation are large enough to compensate for any
GHG-induced increase in PET in the post-industrial period.
**6. Summary and discussion**
This study presents the first comprehensive assessment of past variability and trends in
hydroclimate of northern European sector over the last millennium of the Common Era along
with interrelated variables: precipitation, which supplies moisture, and temperature, which
modulates evapotranspiration. A combined approach comparing observational (both
instrumental and proxy based) and model-based results is used for evaluation of simulated
and real-world interannual-to-centennial climate variability and the underlying physics
governing the climate system. A number of important finding emerge from the collective
comparison:
[1] Models and proxy data are found to broadly agree on the modes of atmospheric variability
(sect. 5) that produces droughts and pluvials in Scandinavia. Despite these dynamical
similarities, the GCMs are, however, not able to reproduce the hydroclimate features in the
proxy record. The droughts and pluvials in the forced simulation are not temporally
synchronous with those in the proxy record, nor do the GCM spectra agree with the proxy
spectra on the amount of variance present on short and long timescales (sect. 3).
[2] The mechanisms that are linking long-term regional summertime moisture variability and
temperature are found to be largely missing in the current generation of models (sect. 5). A
weak negative association between the two components is revealed from observational and
proxy evidences on interannual timescales, while on decadal timescales a positive correlation
dominates. The timescale dependent relationship between regional precipitation and
temperature is considerably biased in the CMIP5/PMIP3 models, which is reflected in the
overestimation of the short-term negative association and significant underestimation of the
long-term relationship between them. This discrepancy is most likely arising from the spectral
inconsistencies among the model and proxy datasets.
[3] There are considerable disagreements among hydroclimate features shown by the
CMIP5/PMIP3 simulations (despite the use of largely similar forcing series) (sect. 3).
Together, these results point to the possibilities of dominant influence of stochastic processes



for the regional hydroclimate and/or deficiencies in the models to realistically represent
relevant processes in reality.

Essentially, our results reveal that the GCM simulated interannual components of the

variability are overestimated, while the multidecadal/longer timescale components are
generally too weak. Earlier studies (Ault et al., 2012; Ault et al., 2013) have also argued that
most CMIP5/PMIP3 models exhibit less hydroclimate persistence than the instrumental or
proxy records. It is difficult to determine explicitly whether it is an external forcing or internal
sources that drive the decadal and longer variance in the proxy reconstruction. Prior studies
have highlighted the importance of external influences on regional climate variability at
different timescales (e.g., Gleckler et al., 2006; Thiéblemont et al., 2015; Sigl et al., 2015).
Although we find a short term response of regional hydroclimate to volcanic perturbations
(sect. 4), multi-year anomalies in the proxy reconstruction do, however, not appear to
correspond well with the epochs following the large volcanic eruptions (e.g., in the 1250s,
1450s and 1810s) used to force the models. Thus we cannot rule out that the variability in the
reconstruction largely arise from internal sources of variation. Consequently, if the proxy-
inferred decadal-to-multidecadal variability is accurate and if the variability is indeed largely
unforced, then its magnitude is well beyond what any of the current generation global climate
models are able to produce in the region. Underestimation of redness in the models on
multidecadal/longer timescales, suggests the GCMs might be lacking/underestimating
processes important to the variability at these timescales. There are a number of recognized
limitations relating to the dynamics that are relevant to the climatology of the North Atlantic-
European sectors. One such example is that models have generally been unable to simulate
low-frequency variability in the North Atlantic Oscillation (Osborn, 2004). They have also
been shown to underestimate the periodicity of the Atlantic Multidecadal Oscillation
(Kavvada et al., 2013), which has implications for the associated hydroclimate impact on
neighboring continents (Coats et al., 2015).  If, on the other hand, the proxy estimated
multidecadal/longer variability in the last millennium is forced by exogenous mechanisms,
then either 1) it is a forcing component that is largely missing in the CMIP5/PMIP3 models,
alternatively, 2) it is a forcing component that generates a different long-term response in the
models compared to the "proxy view" of regional hydroclimatology.

It is not possible to pinpoint which part of the disagreement between models and the

proxy comes from uncertainties in the tree-ring reconstruction, deficiencies in the forcing
series used to drive the models, or from deficiencies in the model. Our analyses have mainly
been based on precipitation simulation – a challenging variable for GCMs to simulate



accurately. The coarse spatial resolution of the models gives only an approximate
representation of the topographic features, which are important for regional hydroclimate.
Another possibility is that the scale of the GCMs is unrepresentative of the point estimate
provided by the ScandH17 reconstruction. On the other hand, the mismatch between grid box
and point estimates is expected to reduce at longer timescales (Jones et al., 1997). There are
also limitations of the tree-ring proxy and uncertainties in the interpretation of the data that
cannot be ignored. Tree-rings and other biological archives may integrate climate conditions
over multiple years (Zhang et al., 2015), which could potentially overestimating the ratio of
low to high frequency variability (Franke et al., 2013). While we have been able to establish
that prevailing summer moisture availability has been the main growth limitation of trees in
the ScandH17 network on an interannual basis over the twentieth century (Figs. S3 and S6),
we cannot verify the drought-tree growth model in the pre-instrumental era or across longer
spectrum of variability. We are not able to rule out that there might have been climatic
regimes in the past that would have caused dynamical shift in the tree growth response to
climate, and potentially have called into question the uniformitarian paradigm traditionally
applied in the field of dendroclimatology. There are risks that less well know dynamics
outside the climate system may introduce variability into the records at decadal/longer
timescales. Advances in the mechanistic understanding of the various proxies and the
processes through which they record environmental change, e.g., through development and
refinement of process-based forward models (Tolwinski-Ward et al., 2011), is currently an
emerging priority in the field.

The discrepancies in CMIP5/PMIP3 simulated and proxy reconstructed hydroclimate

variability in the last millennium is an issue that must be addressed when assessing
projections of future hydroclimate change. The lack of adequate understanding for
mechanisms linking temperature and moisture supply on longer timescales has important
implication for future projections. Weak multidecadal variability in models also implies that
inference about future persistent droughts and pluvials based on the latest generation global
climate models will likely underestimate the true risk of these events. Reconciliations for the
apparent proxy – model mismatch will require efforts from the proxy, modeling and statistics
groups, including additional proxy records and refined model simulations of hydroclimate
variability in the last millennium, together with the development of alternative approaches for
joint proxy-model assessments. Having here provided a first comparison of reconstructed and
simulated hydroclimate for Scandinavia, our future efforts will include adaptions of the data
assimilation approach to paleoclimate reconstruction. Such efforts hold promise for reducing



598 the uncertainties associated with model physics, external forcings, and internal climate
599 variability, and ultimately help to refine our view of past and future hydroclimate changes.

**Data availability**

601 The raw tree-ring data can be downloaded from the International Tree-Ring Data Bank
602 (http://www.ncdc.noaa.gov/paleo/treering.html) and the SAIMA Tree-Ring Data Bank
603 (http://lustiag.pp.fi/Saima/dendrotieto.htm) (Table II). The CMIP5/PMIP3 climate model
604 output can be obtained though the Earth System Grid - Center for Enabling Technologies
605 (ESG-CET) portal (http://pcmdi9.llnl.gov/). The ScandH17 hydroclimate reconstruction is
606 archived through the NOAA paleoclimate database (citation added on publication).

**Acknowledgments**

608 K. Seftigen was supported by the FORMAS mobility starting grant for young researchers
609 (grant # 2014-723). H. Goosse is senior research associate with the FRS/FNRS, Belgium. The
610 authors wish to acknowledge the World Climate Research Programme's Working Group on
611 Coupled Modelling, which is responsible for CMIP, and to thank the climate modeling groups
612 (listed in Table I of this paper) for producing and making available their model output. For
613 CMIP the U.S. Department of Energy's Program for Climate Model Diagnosis and
614 Intercomparison provides coordinating support and led development of software
615 infrastructure in partnership with the Global Organization for Earth System Science Portals.
616 The authors also wish to acknowledge the researchers who have produced and made their
617 tree-ring chronologies available.

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





**Tables and figures**
**Table I.** CMIP5/PMIP3 model description.

| Model Name | Resolution [Atmosphere] | Resolution [Ocean] | Reference |
|---|---|---|---|
| CCSM4 | 192 x 288 | 384 x 320 | Landrum et al. (2012) |
| CESM1 | 96 x 144 | 384 x 320 | Lehner et al. (2015) |
| IPSL-CM5A-LR | 96 x 96 | 149 x 182 | Dufresne et al. (2013) |
| MIROC-ESM | 64 x 128 | 192 x 256 | Watanabe et al. (2011) |
| MPI-ESM-P | 96 x 192 | 220 x 256 | Jungclaus et al. (2014) |
| BCC-CSM1-1 | 64 x 128 | 232 x 360 | Wu et al. (2014) |




**Table II:** Tree-ring network description.

| Site | Coord. | Time coverage | Standardization method | MSL[3] | Source |
|---|---|---|---|---|---|
| Eastern Finland | 61.87N, 28.90E | 535 -2002 CE | RCS[1] | 147 yrs | Helama et al. (2009) Online resource: http://lustiag.pp.fi/Saima/dendrotieto.htm Date access: January 2013 |
| Gotland Sweden | 57.49N, 18.41E | 1127-2011 CE | RCS | 130 yrs | Investigator: Schweingruber, F.H. Online resource: https://www.ncdc.noaa.gov/paleo/study/4427 Date access: January 2013 Updated in Seftigen et al. (2015) |
| Jondalen Norway | 59.71N, 9.53E | 1185 -2011 CE | RCS | 165 yrs | Investigator: Briffa, K. Online resource: https://www.ncdc.noaa.gov/paleo/study/2826 Date access: January 2013 Updated in Seftigen et al. (2015) |
| Baljåsen Sweden | 59.04N, 12.27E | 1686-2002 CE | SF2[2] | 174 yrs | Seftigen et al. (2015) |
| Björbo Sweden | 60.27N, 14.44E | 1450-2011 CE | SF | 177 yrs | Investigator: Axelson, T. Online resource: https://www.ncdc.noaa.gov/paleo/study/2667 Date access: January 2013 |
| Ekhultebergen Sweden | 57.45N, 13.50E | 1705-2008 CE | SF1 | 215 yrs | Seftigen et al. (2015) |
| Fårhagsberget Sweden | 58.08N, 16.14E | 1621-2011 CE | SF1 | 262 yrs | Seftigen et al. (2015) |
| Helvetets håla Sweden | 57.14N, 16.14E | 1691-2011 CE | SF1 | 255 yrs | Seftigen et al. (2015) |
| Halle-Vagnaren Sweden | 57.17N, 15.17E | 1718-2009 CE | SF3 | 186 yrs | Seftigen et al. (2015) |
| Hornslandet Sweden | 59.01N, 11.08E | 1590-2011 CE | SF1 | 270 yrs | Seftigen et al. (2015) |
| Korphålorna Sweden | 61.43N, 17.00E | 1790-2011 CE | SF1 | 199 yrs | Seftigen et al. (2015) |
| Myrkaby Sweden | 57.45N, 15.23E | 1669-2011 CE | SF2 | 294 yrs | Seftigen et al. (2015) |
| Nämndö Sweden | 59.52N, 16.56E | 1582-1995 CE | SF1 | 123 yrs | Investigator: Larsson, L. Online resource: https://www.ncdc.noaa.gov/paleo/study/3869 Date access: January 2013 |
| Valekleven-Ombo Sweden | 59.11N, 18.41E | 1578-2011 CE | SF1 | 225 yrs | Seftigen et al. (2015) |
| Putbergen Sweden | 58.37N, 14.32E | 1734-2008 CE | SF1 | 188 yrs | Seftigen et al. (2015) |
| Salboknös Sweden | 59.11N, 16.55E | 1486-2011 CE | SF2 | 357 yrs | Seftigen et al. (2015) |
| Särö Sweden | 61.92N, 11.93E | 1712-2002 CE | SF3 | 176 yrs | Seftigen et al. (2015) |
| Sisshammer Sweden | 59.46N, 14.54E | 1661-2003 CE | SF | 74 yrs | Investigator: Andreason, T. Online resource: https://www.ncdc.noaa.gov/paleo/study/2663 Date access: January 2013 |
| Skärmarbodabergen Sweden | 57.51N, 11.93E | 1600-2002 CE | SF3 | 160 yrs | Seftigen et al. (2015) |
| Skitåsen Sweden | 59.09N, 18.02E | 1672-2011 CE | SF2 | 285 yrs | Seftigen et al. (2015) |
| Skuleskogen Sweden | 59.26N, 15.07E | 1448-2011 CE | SF | 181 yrs | Seftigen et al. (2015) |
| Sörknatten Sweden | 59.22N, 15.29E | 1762-2009 CE | SF3 | 197 yrs | Seftigen et al. (2015) |
| Tjurhults mosse Sweden | 63.06N, 18.29E | 1655-2011 CE | SF2 | 268 yrs | Seftigen et al. (2015) |
| Tjusthult Sweden | 58.55N, 12.27E | 1681-2011 CE | SF1 | 221 yrs | Seftigen et al. (2015) |
| Tyresta Sweden | 59.52N, 14.71E | 1609-2010 CE | SF1 | 198 yrs | Linderholm and Molin (2005) Updated in Seftigen et al. (2015) |

[1] RCS: Regional Curve Standardization;
[2] SF: Signal-Free Standardization. The number after the abbreviation indicates the PCA cluster number (Fig. S2);
[3] MSL: Mean Segment Length.





**Table III.** Event years used in the Superposed Epoch Analysis (Fig. 4). The event lists are composed
of the 20 strongest eruptions from each record.

| Source | Event years (CE) |
|---|---|
| Gao et al. (2008) (sulfate aerosol > 15 Tg) | 1167, 1176, 1195, 1227, 1258, 1284, 1328, 1452, 1459, 1584, 1600, 1641, 1719, 1783, 1809, 1815, 1831, 1835, 1991 |
| Crowley and Unterman (2013) (AOD > 0.13) | 1229, 1258, 1259, 1286, 1287, 1456, 1457, 1600, 1601, 1641, 1695, 1696, 1809, 1810, 1815, 1816, 1817, 1884, 1992 |
| Sigl et al. (2015) (global forcing < 5.86 W/m$^2$) | 1108, 1171, 1191, 1230, 1258, 1276, 1286, 1345, 1453, 1458, 1601, 1641, 1695, 1783, 1809, 1815, 1832, 1836, 1992 |



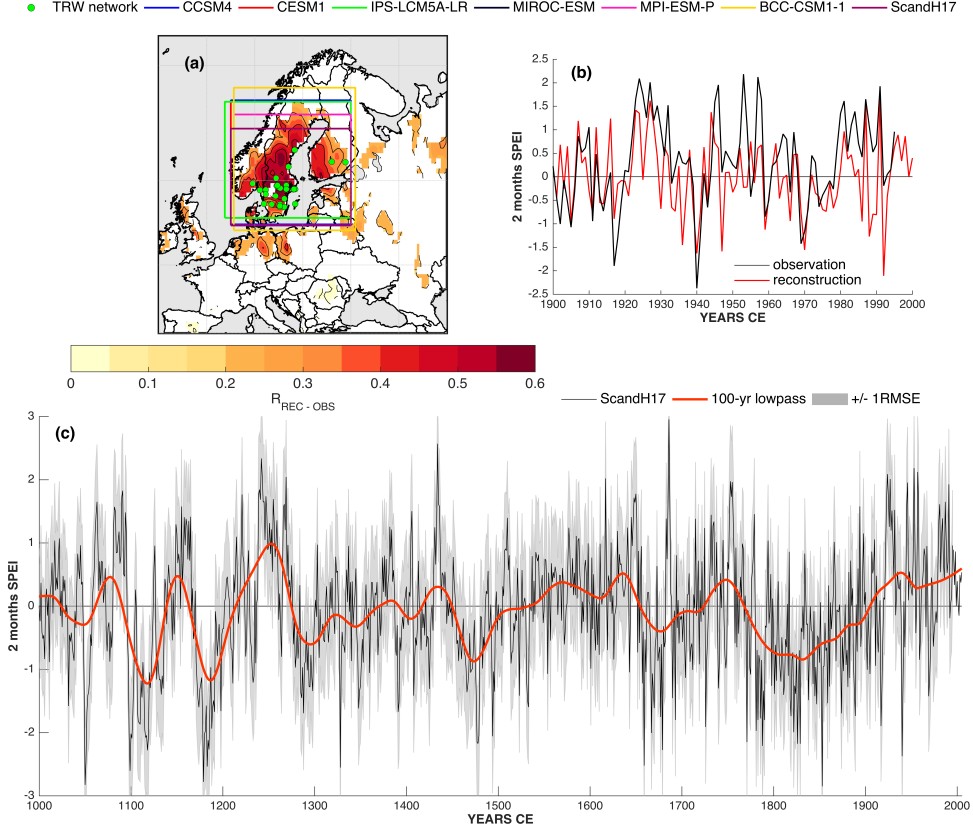

**Figure 1: Average regional SPEI time series reconstructed from tree-rings.** (a) Location of the tree-ring network used for regional reconstruction and the extent of the CMIP5/PMIP3 model precipitation and temperature grids used to derive regional SPEI estimates. Shaded contours display the correlation (p < 0.1) between the tree-ring reconstruction and fields of instrumental SPEI data over the 1901-1995 period; (b) average regional reconstructed and instrumental 20[th] century 2-month June SPEI; (c) average regional SPEI nested reconstruction, with the +/- 1RMSE of the regression equations outlined in grey shading. A smoothed version of the reconstruction using a 100-year loess smooth is shown in red. Reconstruction assessment metrics are provided in supplementary materials (Fig. S4).





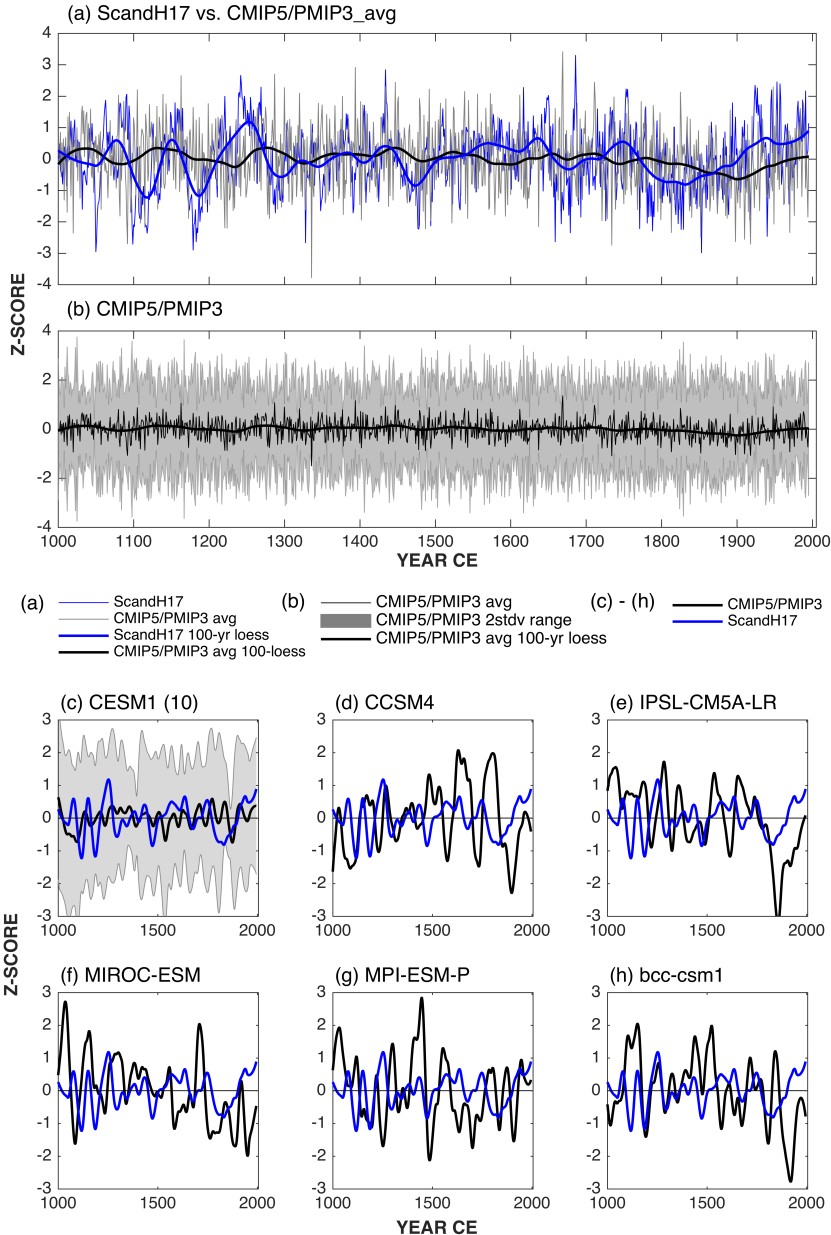

**Figure 2: Comparison of reconstructed SPEI with forced model runs.** (a) The reconstruction versus the mean of the six CMIP5/PMIP3 models transformed into standard normal deviates (z-scores) over the 1000-1995 CE period and smoothed with 100-year loess filter; (b) multimodel mean and the two standard deviation range (shading) of the six GCMs; (c) mean and two standard deviation (shading) of CESM1 ten smoothed and z-scored ensemble members (blue) together; (d) – (h) the



reconstruction (blue) versus individual model runs (black). All time series have been smoothed with 100-year loess filter and then z-scored over the 1000-1995 CE period.

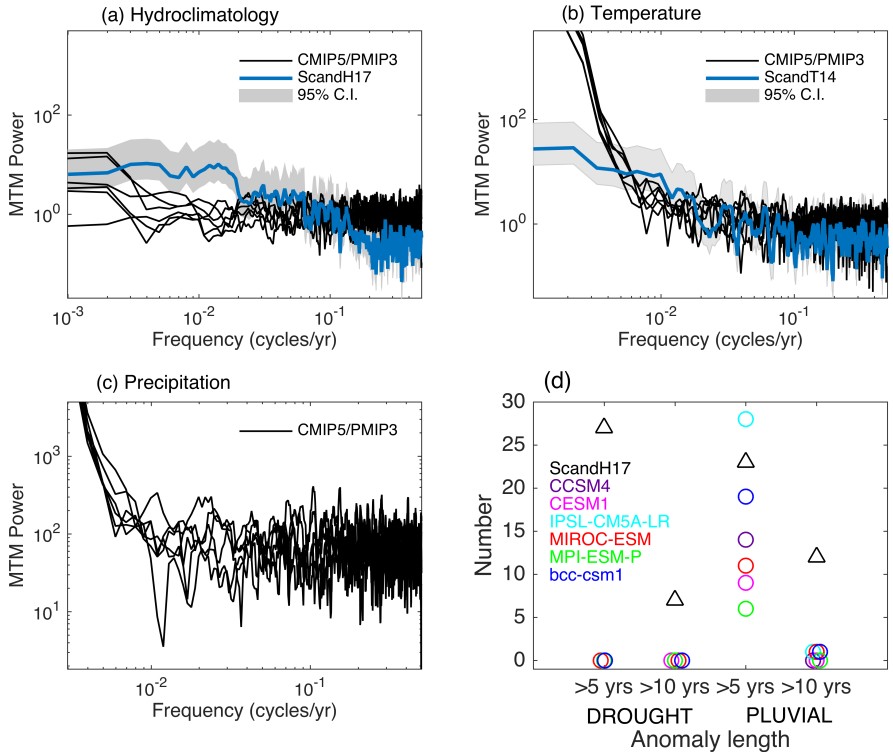

**Figure 3:** Spectral properties (multi-taper approach, 4 tapers) of (a) SPEI, (b) temperature and (c) precipitation over the common 1100-1995 period. For SPEI and temperature, the spectral properties of individual GCMs (r1i1p1 ensemble) are compared to those of the tree-ring ScandH17 and ScandT14 reconstructions. Shaded areas show the 95% confidence interval of the reconstruction spectra. (d) The number of droughts and pluvials in the reconstructed and simulated time series that are > 5 and >10 years in duration. Spectral properties of the individual models are provided in Fig. S7.



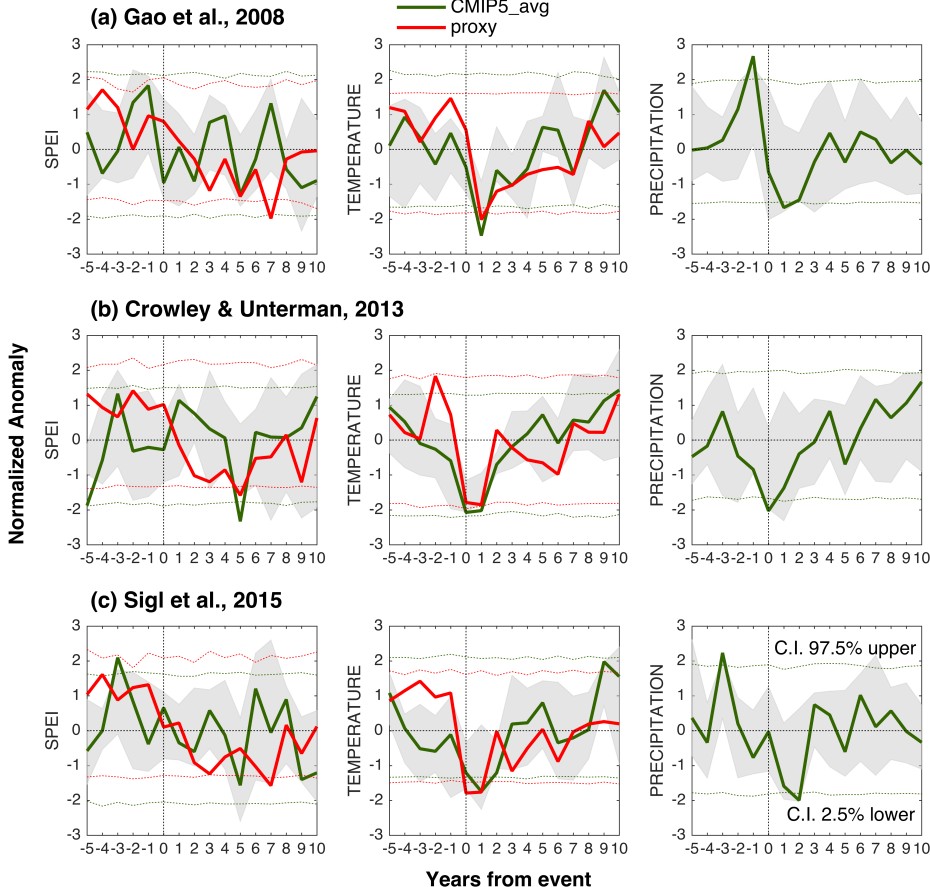

**Figure 4: Modeled and reconstructed hydroclimate response to eruptions.** Superposed epoch analysis using the 20 largest eruption years from the (a) Gao et al. (2008), (b) Crowley and Unterman (2013), and (c) Sigl et al. (2015). Table III lists the event years used in the analysis. Grey shading indicate the range of modeled hydroclimate response from the six GCMs. Confidence intervals (C.I.) are derived from bootstrap resampling (N = 10 000).




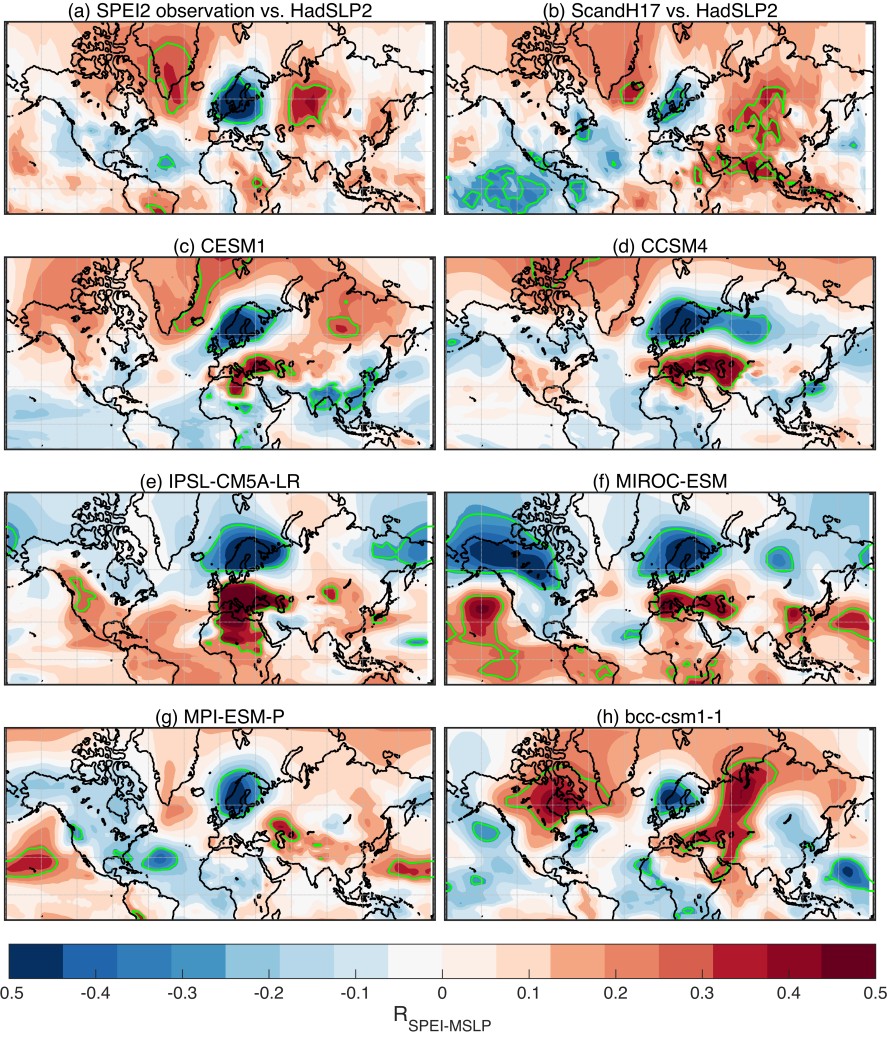

**Figure 5: Spatial distribution of correlation coefficient of northern European warm season hydroclimate and mean sea level pressure (MSLP).** Association between regional drought index and sea level pressure over the 1950-1995 period. (a) observational, (b) ScandH17, (c)-(g) model based results (including r1i1p1 ensemble only). Regions with significant ($p < 0.05$) correlations are outlined in green contours.



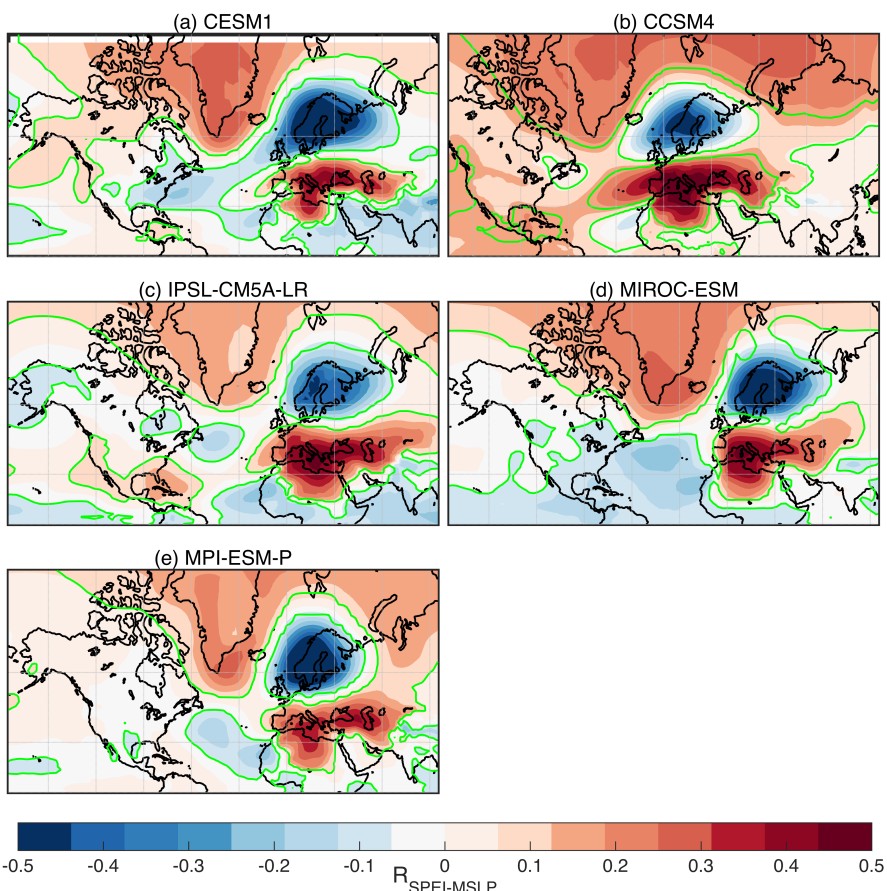

**Figure 6: Spatial distribution of correlation coefficient of northern European warm season hydroclimate and mean sea level pressure (MSLP).** Same as Fig. 5, but for the 850-1849 CE period.





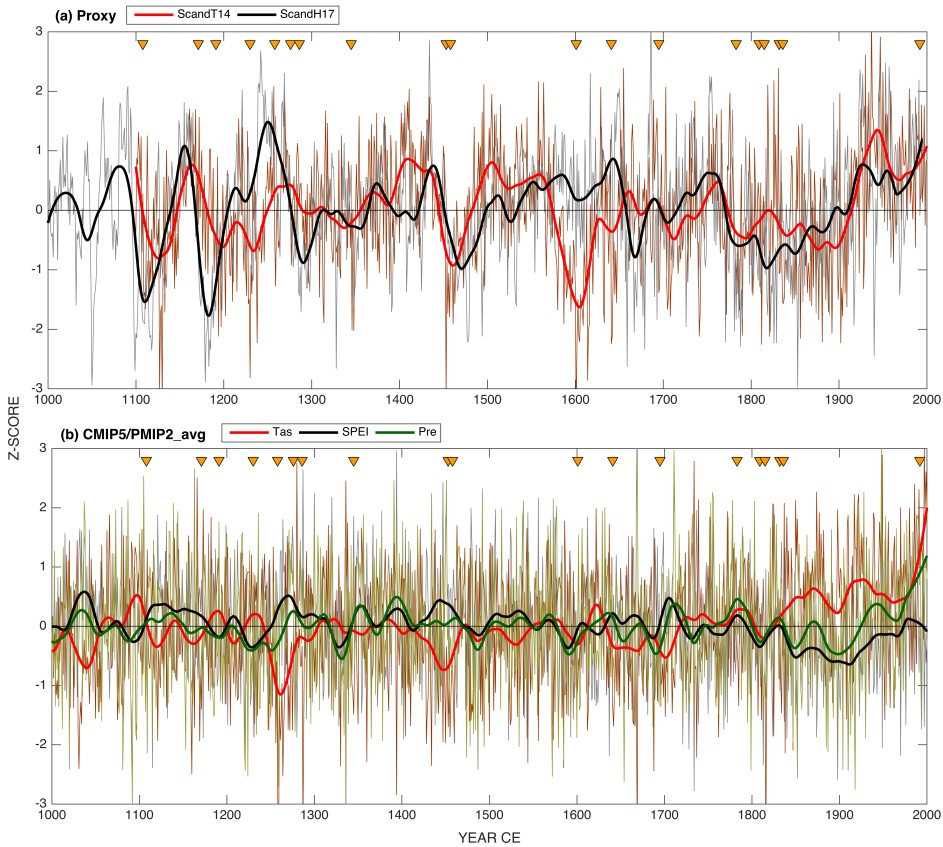

**Figure 7:** Time series of (a) ScandH17 and ScandT14 , and (b) GCM (r1i1p1 ensemble) average temperature, precipitation and SPEI. Smoothed time-series using a 50-year loess filter are shown as thick lines. Individual model data are provided in supplementary material (Fig. S8). The years with volcanic eruptions from Table III are indicated by triangle glyphs.





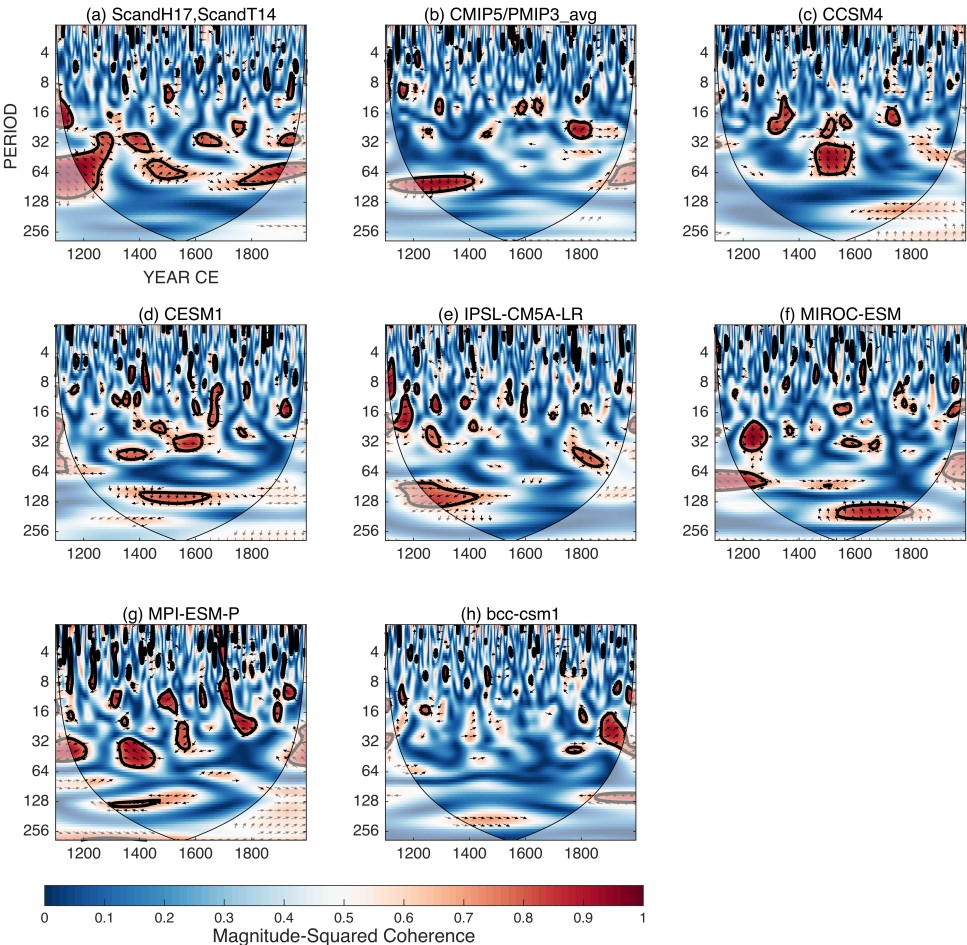

**Figure 8:** Squared wavelet coherence and phase between (a) ScandH17 and ScandT14, and (b) – (h) CMIP5/PMIP3 simulations of temperature and rainfall. The arrows indicate the relative phase relationship between two series; right (left) pointing arrow indicates in-phase (180 degrees out of phase) relationship. Significant coherence at 95% significance level is shown as thick contour.