# Peer review of "Discussion started: 5 April 2017"

_Climate of the Past, 2017_

## Referee Comment (RC1) · Anonymous Referee #1 · 30 May 2017

The manuscript presents a novel reconstruction of soil moisture availability for Southern Scandinavia and compares it with CMIP/PMIP models and a regional temperature reconstruction. The key questions are, if the two proxy reconstructions and the GCM output show similar climate variability and patterns and to investigate possible drivers of hydroclimate variability in the study regions. While general patterns appear to be similar, there are considerable mismatches at most time scales and the authors raise the issue, that multi-decadal hydrological variability might be underrepresented in the GCMs, with substantial impact on future projections of hydrological extremes.

The paper is well-written, comprehensive and I recommend it to be published. It raises important questions about the ability of GCMs to represent regional hydroclimate cor-

rectly and the discrepancy between proxy reconstructions and GCM output is a crucial problem to assess the performance of both reconstructions of hydroclimate as well as GCMs to predict future developments.

In general I find some formulations too strong and not fully supported by the presented analysis. This is in particular the case for formulations of over/underestimation and biases in the model output in comparison to the SPEI reconstruction. When comparing two non-perfect representations of a variable these terms are in my view only justified if one also includes observational data or if there is compelling evidence that one of the two is a better representation of physical reality. At most parts of the manuscript suggests that the proxy reconstruction is an accurate representation and the discrepancy is mainly due to an inability of the GCMs to reproduce these features. A better way would be, to just state a difference in variability, possibly followed by an assessment of both representations. The authors state in lines 561-562, that it is not possible to attribute the disagreements between the reconstruction and the models to one side. This is in disagreement with the rest of the paper, which blames the models. A reasonable formulation is found in lines 544-547 and I would have liked to see similar remarks earlier in the text.

This can probably be easily resolved by reformulating certain statements. In the current form, I found it widely irritating while reading. Some examples of statements which I find too strong (there might be others) are:

- l. 19-20, l. 315-318, l. 460-465 and l. 532-533: I would like to see a critical assessment of the variability of the SPEI reconstruction and a deeper comparison to the expected time-scale depended variability, also from observational data. Just comparing the model output to the reconstruction is in my view not enough to conclude a bias on the model-side.

- l. 22-23, l. 520-522: The formulation implies, that a positive correlation on multidecadal scales is also found from observational evidence. This is not provided in the

manuscript, but is crucial to determine if the mismatch between proxies and models is mainly due to GCMs deficiencies. Thus, much of the paragraph l. 518-526 is formulated with a bias towards a correct representation in the proxy reconstruction. The fact, that the multi-decadal variability is much stronger in the reconstructions can have many reasons (some of which are also discussed in the manuscript), but to claim a bias in the models from this fact alone is a bit strong.

Specific Questions

- l. 215-216: It is not immediately clear, which period is the validation and which is the calibration period.

- Fig 3: It would be nice to include (at least for the high-frequency part) observations into this plot, to be able to judge both the proxy and the model performance. In Fig. 3 (d), adding the markers to the legend would make it more intuitive to read, especially for monochromatic prints.

- Fig 4.: I found it a bit confusing that the order of the columns does not correspond to the order in which the variables are discussed in the text. I would recommend adjusting the order accordingly.

- Repeatedly the formulations seem to imply that hydroclimate and temperature/precipitation are independent variables which one can "contrast" or "compared" (e.g. l. 227). While the first is rather a combination of the two and thus one is not comparing them, but rather investigating, which factor is more dominating.

- As a reader who is not familiar with SPEI it is hard to follow what this variable does and what it's dependencies are. I would have liked to see the formula that is used in this study, possibly in the Supplementary Material.

- l. 235-236: While the two time series are coming from different data sets they might still share common signals and might not be totally independent. Given the rather low $r^2$ of 0.2, one could also argue that the low-frequency variability of the SPEI index, which is a combination of precipitation and temperature, is simply dominated by the temperature component, which would lead to similar multi-decadal variability with ScandT14.

- l. 309/310: A reference to the Supplementary Material Sec. S1 could strengthen this claim, even though it only applies to the inter-annual time scale.

- Sec. 4: I found the title misleading, as the only external forcing discussed is volcanic eruptions, while other forcings like solar variability are not mentioned. Please revise the title.

- l. 432 ff: I found this paragraph a bit confusing. While it begins with comparison of temperature and precipitation the results are about temperature and SPEI. As SPEI is a mixed variable, which also includes temperature, it is not clear, how one can draw a connection to a T-precip relationship. In general, it seems like SPEI and precipitation are used interchangeable here, which they are not.

- l. 448 ff: Can you quantify the relationship between ScandH17 and ScandT14? While the multi-decadal co-variability is clearly visible by eye this is not the case for the inter-annual values. In both cases it would be nice to have quantified values (including significance).

- Sec./Fig. S1: I'm a bit lost here. How can the annual mean be overestimated if all monthly means are underestimated?

Technical Corrections

- 387 [. . .] superior [to] TRW [. . .]

―――――――――――――――――

---

## Short Comment (SC1) · 3 Jun 2017

The PAGES Data Stewardship Integrative Activity seeks to advance best practices for sharing data generated and assembled as part of all PAGES-related activities. As part of this activity, a team of reviewers has been constituted for the "Climate of the Past 2000 years" Special Issue. The data team is reviewing the data handling within each of the CP-Discussion papers in relation to the CP data policy and current best practices. The team has identified essential and recommended additions for each paper, with the goal of achieving a high and consistent level of data stewardship across the 2k Special Issue. We recognize that an additional effort will likely be required to

meet the high level of data stewardship envisaged, and we appreciate the dedication and contribution of the authors. This includes the use of Data Citations (see example in supplement). We ask authors to respond to our comments as part of the regular open interactive discussion. If you have any questions about PAGES Data Stewardship principles, please contact any of us directly.

Best wishes for the success of your paper,

2k Special Issue Data Review Team (Darrell Kaufman, Nerilie Abram, Belen Martrat, Raphael Neukom, Scott St. George) and ex-officio team members (Marie-France Loutre, Lucien von Gunten)

Essential additions for this paper:

(1) Expand the "Data Availability" section to include a Data Citation or URLs to the primary output of this study (regional SPEI nested reconstruction (ScandH17) and the 100-year smooth and estimate uncertainty).

(2) Add Data Citations or URLs (in addition to publication citations) for each of the 27 tree-ring chronologies used in this study to Table II (we note that Table II includes only 25 entires). For those raw data not already in a persistent public repository, submit the essential metadata along with the chronology itself and add the corresponding Data Citation (or URL) in Table II. The archived data must contain the modified chronologies as they were re-processed and used in this study (newest signal-free standardization; adjusted to reduce variance bias). The 'Updated by Seftigen et al. 2015' revisions should be publicly archived and the 'Seftigen et al. 2015' datasets should also be archived.

(3) Add a Data Citation for the ScandT14 reconstruction (Fig 7a). If the data have not previously been deposited in a public data repository, then submit the essential metadata along with the time series itself and add the corresponding Data Citation and publication citation to the caption for Fig 7.

We also suggest the following additions:

(4) Archive the target time series for the reconstruction (20th century 2-month June SPEI, Fig 1b), or at least include the URL link to the CRU data that underpins the time series.

(5) Archive the mean of the six GCM simulations for the last millennium, including the SPEI (Fig 2a, 7b) and temperature and precipitation (Fig 7b). Archiving the model mean data will assist others in carrying out future data-model comparisons as new tree-ring chronologies are published.

Please also note the supplement to this comment:
http://www.clim-past-discuss.net/cp-2017-36/cp-2017-36-SC1-supplement.pdf

[Figure]

**Supplement:**

Data Citations track the provenance of a dataset giving credit to the data generator; this is in addition to any references to publications where the data are described. Data Citations are used in the text (or tables) alongside and in the same way as publication citations. In the Reference list, they include: Creators, Title, Repository, Identifier, Submission Year. More information about Data Citations is here: <https://www.datacite.org/mission.html>

Here is an example of text and corresponding citations (using CP punctuation style):

The PAGES2k Consortium (in press) assembled a large global dataset of temperature-sensitive proxy records (PAGES2k Consortium, 2017). Among the records is the paleo-temperature reconstruction from Laguna Chepical (de Jong et al., 2016), which was described by de Jong et al. (2013).

de Jong, R., von Gunten, l., Maldonado, A., and Grosjean, M.: Late Holocene summer temperatures in the central Andes reconstructed from the sediments of high-elevation Laguna Chepical, Chile (32° S), Climate of the Past, 9, 1921-1932, 2013.

de Jong, R., von Gunten, l., Maldonado, A., and Grosjean, M.: Laguna Chepical summer temperature reconstruction, World Data Center for Paleoclimatology, https://www.ncdc.noaa.gov/paleo/study/20366, 2016.

PAGES 2k Consortium: A global multiproxy database for temperature reconstructions of the Common Era, Scientific Data, in press.

PAGES 2k Consortium: A global multiproxy database for temperature reconstructions of the Common Era, version 2.0.0, figshare, https://figshare.com/s/d327a0367bb908a4c4f2, 2017.

---

## Referee Comment (RC2) · Anonymous Referee #2 · 14 Jul 2017

This paper generated a new field hydrological reconstruction over Scandinavia covering the last millennium based on a network of a few dendro-chronologies. The authors mainly compared this reconstruction with a few selected simulations from CMIP5 and PMIP3. The article is well written and prepared. I have only a few comments, as following: 1. The growth of tree-ring width (TRW) may be limited by the shortage of water. However, does the TRW positively and linearly depend on soil mositure/ precipitation? Can TRW proxies reflect floods/extreme wetness, especially when the study region is not arid? 2. I see you have another field hydrological reconstruction over Fennoscandia based on much more proxy records. Have you compared this reconstruction with that one over Scandinavia? Is there any difference for the northern part? 3. The comparison between the reconstruction and the simulations is interesting. However some conclusions, from my point of view, are too strong. For example, "We find simulated interannual components of variability to be overestimated, while the multidecadal/longer timescale components generally are too weak." I supposed the conclusion is drawn from the lines from 307-322. As far as I understand, the TRWs tend to have red biased spectra, please see the papers from Franke et al. (2013) and Bunde et al. (2013). So, is it possible that the TRW-based reconstruction overestimated low-frequencies? If that is the case, then the following conclusions are not solid. Especially," Weak multidecadal variability in models also implies that inference about future persistent droughts and pluvials based on the latest generation global climate models will likely underestimate the true risk of these events."

References: Bunde A, Buentgen U, Ludescher J, Luterbacher J and von Storch H (2013) Is there memory in precipitation? Nat. Clim. Change 174–5 Franke J, Frank D, RaibleCC, Esper J and Broennimann S (2013) Spectral biases in tree-ring climate proxies Nat. Clim. Change 360–4

---

## Author Comment (AC1) · 15 Sep 2017

*We would like to acknowledge the time and effort that Editor Dr. Helen McGregor, 2k Special Issue Data Review Team, and the two anonymous reviewers have put into assessing the previous version of the manuscript. We have provided the responses to each of the reviewers' comments below. These are shown in blue italics.*

*We look forward to hearing from you in due time regarding our resubmission and to respond to any further questions and comments you may have.*

*On behalf of the authors,*

*Kristina Seftigen*

**Referee #1**
In general I find some formulations too strong and not fully supported by the presented analysis. This is in particular the case for formulations of over/underestimation and biases in the model output in comparison to the SPEI reconstruction. When comparing two non-perfect representations of a variable these terms are in my view only justified if one also includes observational data or if there is compelling evidence that one of the two is a better representation of physical reality. At most parts of the manuscript suggests that the proxy reconstruction is an accurate representation and the discrepancy is mainly due to an inability of the GCMs to reproduce these features. A better way would be, to just state a difference in variability, possibly followed by an assessment of both representations. The authors state in lines 561-562, that it is not possible to attribute the disagreements between the reconstruction and the models to one side. This is in disagreement with the rest of the paper, which blames the models. A reasonable formulation is found in lines 544-547 and I would have liked to see similar remarks earlier in the text.

This can probably be easily resolved by reformulating certain statements. In the current form, I found it widely irritating while reading. Some examples of statements which I find too strong (there might be others) are:

- l. 19-20, l. 315-318, l. 460-465 and l. 532-533: I would like to see a critical assessment of the variability of the SPEI reconstruction and a deeper comparison to the expected time-scale depended variability, also from observational data. Just comparing the model output to the reconstruction is in my view not enough to conclude a bias on the model-side.

- l. 22-23, l. 520-522: The formulation implies, that a positive correlation on multidecadal scales is also found from observational evidence. This is not provided in the manuscript, but is crucial to determine if the mismatch between proxies and models is mainly due to GCMs deficiencies. Thus, much of the paragraph l. 518-526 is formulated with a bias towards a correct representation in the proxy reconstruction. The fact, that the multi-decadal variability is much stronger in the reconstructions can have many reasons (some of which are also discussed in the manuscript), but to claim a bias in the models from this fact alone is a bit strong.

*Response: we agree with the reviewer that some of the statements are perhaps formulated too strong. Following the reviewer suggestions we have now reformulated some of the text. Rather than attributing the mismatch to either of the datasets, sect. 3 and 5 are now more focused on describing the inconsistencies in the model and proxy records. Possible causes of the disagreement are briefly discussed in the final sect. 'summary and discussions'. The*

*abstract has been rephrased accordingly.*

- l. 215-216: It is not immediately clear, which period is the validation and which is the calibration period.

*Response: We have used a split period calibration/validation procedure, which means that both the early (1901-1948) and the late (1949-1995) periods have been used for calibration and validation. A more detailed description of the calibration/validation scheme (l.198-202) is now provided.*

- Fig 3: It would be nice to include (at least for the high-frequency part) observations into this plot, to be able to judge both the proxy and the model performance. In Fig. 3 (d), adding the markers to the legend would make it more intuitive to read, especially for monochromatic prints.

*Response: Thank you for the suggestion. We have now provided observational data in the comparison. Also, the legend in fig. 3d is changed as suggested.*

- Fig 4.: I found it a bit confusing that the order of the columns does not correspond to the order in which the variables are discussed in the text. I would recommend adjusting the order accordingly.

*Response: Order revised*

- Repeatedly the formulations seem to imply that hydroclimate and temperature/ precipitation are independent variables which one can "contrast" or "compared" (e.g. l. 227). While the first is rather a combination of the two and thus one is not comparing them, but rather investigating, which factor is more dominating.

*Response: Agreed, the text is now revised.*

- As a reader who is not familiar with SPEI it is hard to follow what this variable does and what it's dependencies are. I would have liked to see the formula that is used in this study, possibly in the Supplementary Material.

*Response: Thank your for this suggestion. A more detailed description of the SPEI computation, as well as some technical notes on the use of the SPEI R package, are now provided in the supplementary materials.*

- l. 235-236: While the two time series are coming from different data sets they might still share common signals and might not be totally independent. Given the rather low r2 of 0.2, one could also argue that the low-frequency variability of the SPEI index, which is a combination of precipitation and temperature, is simply dominated by the temperature component, which would lead to similar multi-decadal variability with ScandT14.

*Response: We thank the reviewer for his/her comment. We however argue that one would expect the two reconstructions ScandH17 and ScandT14 to be anti-correlated in the decadal/lower frequencies (warm decades -> dry decades), would the low-frequency portion of the reconstructed SPEI series be dominated by a temperature component. Interestingly, this is not the case. We have showed in the manuscript that a precipitation signal dominates the*

*high-frequency portion of the TRW variability. Theoretically one would therefore expect the decadal variability to also be driven by changes in rainfall rather than temperature. However, we cannot rule out that there might be a frequency dependent sensitivity of the proxy data to climate and that the influence of temperature could increase towards lower frequencies of the spectra. We have now discussed this issue in lines 581-584.*

- l. 309/310: A reference to the Supplementary Material Sec. S1 could strengthen this claim, even though it only applies to the inter-annual time scale.
*Response: done.*

- Sec. 4: I found the title misleading, as the only external forcing discussed is volcanic eruptions, while other forcings like solar variability are not mentioned. Please revise the title.

*Reply: the header is now revised to 'The role of volcanic forcing'*

- l. 432 ff: I found this paragraph a bit confusing. While it begins with comparison of temperature and precipitation the results are about temperature and SPEI. As SPEI is a mixed variable, which also includes temperature, it is not clear, how one can draw a connection to a T-precip relationship. In general, it seems like SPEI and precipitation are used interchangeable here, which they are not.

*Response: The SPEI and precipitation have here been used interchangeable because there is presently no high-resolution, fully independent proxy reconstruction of rainfall available for the region. Also, we show that the SPEI variability is dominated by a precipitation signal in the region (Fig. S5). However, we agree that some of the readership might find this section a bit confusing. We have therefore revised the text (l444-467), to be more focused on the role of temperature in regional hydroclimate. We have also added two new plots to the supplementary materials (Fig. S9), that are more closely exploring the association between simulated SPEI, temperatures and rainfall across different frequency bands.*

- l. 448 ff: Can you quantify the relationship between ScandH17 and ScandT14? While the multi-decadal co-variability is clearly visible by eye this is not the case for the interannual values. In both cases it would be nice to have quantified values (including significance).

*Reply: done (l.453 and l.458).*

- Sec./Fig. S1: I'm a bit lost here. How can the annual mean be overestimated if all monthly means are underestimated?
*Reply: We are not sure that we understand the referees' comment. Fig S1b is based on model data from southern Scandinavia, where many of the models underestimate the annual rainfall (blue areas in fig. S1a).*

Technical Corrections
- 387 [. . .] superior [to] TRW [. . .]

*Reply: corrected*

**Referee #2**

1. The growth of tree-ring width (TRW) may be limited by the shortage of water. However, does the TRW positively and linearly depend on soil mosiure/ precipitation? Can TRW

proxies reflect floods/extreme wetness, especially when the study region is not arid?

*Response: We thank the reviewer for taking time to review our manuscript and providing us with a number of valuable and insightful comments and suggestions. We understand that the use of TRW data from moist and cool Scandinavia in moisture/rainfall reconstructions might appear controversial to some readers. After all, the developments of moisture sensitive TRW data have traditionally been restricted to lower latitude arid and semi-arid regions, with only a few exceptions for the northern European sector. Yet, we argue that moisture stressed trees do grow in these high-latitude environments and tree-ring chronologies with moisture sensitivity can be developed, at least if species and sites are carefully selected, which has also been proven by a number of recent studies (Helama & Lindholm, 2003; Wilson et al., 2012; Cook et al., 2015 etc.). In our study, we have almost exclusively used TRW data from sites with shallow and well-drained soils - that is, sites where the growth of vegetation is clearly affected by the amount of available moisture. We have therefore been able to build chronologies that are positively and more strongly correlated with moisture availability than with temperature variability, which is demonstrated in figs S3 and S6 in the supplementary materials.*

2. I see you have another field hydrological reconstruction over Fennoscandia based on much more proxy records. Have you compared this reconstruction with that one over Scandinavia? Is there any difference for the northern part?

*Response: yes, in Seftigen et al. (2014) we have used a denser tree-ring network to provide a field reconstruction over much of Fennoscandia. In that study the tree-ring data from the northernmost parts of Fennoscandia were mostly negatively correlated with available moisture and therefore we used it indirectly to reconstruct soil moisture availability, by considering the importance of surface temperature in determining the land surface heat flux, evapotranspiration and consequently the water balance. Mixing tree-ring data with different signals for reconstruction purposes is however not always straightforward, and as the main aim of the current study was to provide as robust and reliable reconstruction for the region we decided to only retain tree-ring data from moisture stressed sites, where the tree-growth is positively related to drought. As is shown in the manuscript fig 1, these sites are mainly located in southern Sweden. Comparing the current reconstruction with the 2014 reconstruction would yield basically the same results over southern Sweden, as the two reconstructions share more or less the same set of predictors in this region. The only difference would be that the current reconstruction contains more low-frequency information than the 2014 reconstruction (see sect. 2.2 in the manuscript). We have not compared the new reconstruction with the previous one over northern Sweden, simply because the signal of the new reconstruction is confined to southern and central Scandinavia (see fig. 1).*

3. The comparison between the reconstruction and the simulations is interesting. However some conclusions, from my point of view, are too strong. For example, "We find simulated interannual components of variability to be overestimated, while the multidecadal/longer timescale components generally are too weak." I supposed the conclusion is drawn from the lines from 307-322. As far as I understand, the TRWs tend to have red biased spectra, please see the papers from Franke et al. (2013) and Bunde et al. (2013). So, is it possible that the TRW-based reconstruction overestimated low-frequencies? If that is the case, then the following conclusions are not solid. Especially," Weak multidecadal variability in models also implies that inference about future persistent droughts and pluvials based on the latest generation global climate models will likely underestimate the true risk of these events."

*Response: We agree that some of the statements were made a bit too strong – it is true that we are currently unable to identify the precise origin of the mismatch. We have now revised the*

*manuscript to be more focused on highlighting the discrepancies in the datasets rather than drawing any conclusion about the source for the mismatch (see response to referee #1).*

**2k Special Issue Data Review Team**
(1) Expand the "Data Availability" section to include a Data Citation or URLs to the primary output of this study (regional SPEI nested reconstruction (ScandH17) and the 100-year smooth and estimate uncertainty).

(2) Add Data Citations or URLs (in addition to publication citations) for each of the 27 tree-ring chronologies used in this study to Table II (we note that Table II includes only 25 entires). For those raw data not already in a persistent public repository, submit the essential metadata along with the chronology itself and add the corresponding Data Citation (or URL) in Table II. The archived data must contain the modified chronologies as they were re-processed and used in this study (newest signal-free standardization; adjusted to reduce variance bias). The 'Updated by Seftigen et al. 2015' revisions should be publicly archived and the 'Seftigen et al. 2015' datasets should also be archived.

(3) Add a Data Citation for the ScandT14 reconstruction (Fig 7a). If the data have not previously been deposited in a public data repository, then submit the essential metadata along with the time series itself and add the corresponding Data Citation and publication citation to the caption for Fig 7.
*Response: The ScandT14 reconstructions will be made available through the NOAA paleoclimate database, and citation will be added to the paper. Metadata, including all new chronologies, re-processed chronologies as well as the new ScandH17 reconstruction (smoothed and raw), will be added to supplementary materials.*

References:
Cook, E.R., Seager, R., Kushnir, Y., Briffa, K.R., Büntgen, U., Frank, D., Krusic, P.J., Tegel, W., van der Schrier, G., Andreu-Hayles, L., Baillie, M., Baittinger, C., Bleicher, N., Bonde, N., Brown, D., Carrer, M., Cooper, R., Čufar, K., Dittmar, C., Esper, J., Griggs, C., Gunnarson, B., Günther, B., Gutierrez, E., Haneca, K., Helama, S., Herzig, F., Heussner, K.-U., Hofmann, J., Janda, P., Kontic, R., Köse, N., Kyncl, T., Levanič, T., Linderholm, H., Manning, S., Melvin, T.M., Miles, D., Neuwirth, B., Nicolussi, K., Nola, P., Panayotov, M., Popa, I., Rothe, A., Seftigen, K., Seim, A., Svarva, H., Svoboda, M., Thun, T., Timonen, M., Touchan, R., Trotsiuk, V., Trouet, V., Walder, F., Ważny, T., Wilson, R. & Zang, C. (2015) Old World megadroughts and pluvials during the Common Era. *Science Advances*, **1**
Helama, S. & Lindholm, M. (2003) Droughts and rainfall in south eastern Finland since AD 874, inferred from Scots pine tree-rings. *Boreal Environment Research*, **8**, 171-183.
Seftigen, K., Björklund, J., Cook, E.R. & Linderholm, H.W. (2014) A tree-ring field reconstruction of Fennoscandian summer hydroclimate variability for the last millennium. *Climate Dynamics*, **44**, 3141-3154.
Wilson, R., Miles, D., Loader, N.J., Melvin, T., Cunningham, L., Cooper, R. & Briffa, K. (2012) A millennial long March–July precipitation reconstruction for southern-central England. *Climate Dynamics*,

---

## Author Response (AR2)

1. Please insert the NOAA paleoclimate data URL for the ScandT14 temperature reconstruction into the Data Availability section.

*Response: Done*

2. As far as I can see the ScandH17 reconstruction and standardized tree-ring chronologies (new and reporcessed) are not in the supplementary material. ScandH17 and the standardized tree-ring chronologies need to be archived and available. I recommend this be done with the NOAA paleoclimate database. As for point 1) above the URL can be inserted in the Data Availability section.

*Response: The ScandH17 reconstruction and all 25 standardized tree-ring chronologies are uploaded as supplementary material.*

3. I am not satisfied that the Seftigen et al. 2015 data, updated or otherwise, have been publicly archived. Table II refers to datasets 'Updated by Seftigen et al. 2015' or 'Seftigen et al. 2015'. I tracked back to Seftigen et al. 2015 to look at these data -- note the correct doi is 10.1175/JCLI-D-13-00734.1 (dash missing in the citation in the References section) -- and the supplementary material associated with Seftigen et al. 2015 only contains tree ring location information. Please comply with the 2k Special Issue Data Review team request "(2) Add Data Citations or URLs (in addition to publication citations) for each of the 27[25] tree ring chronologies used in this study to Table II (we note that Table II includes only 25 entires). For those raw data not already in a persistent public repository, submit the essential metadata along with the chronology itself and add the corresponding Data Citation (or URL) in Table II. The archived data must contain the modified chronologies as they were re-processed and used in this study (newest signal-free standardization; adjusted to reduce variance bias). The 'Updated by Seftigen et al. 2015' revisions should be publicly archived and the 'Seftigen et al. 2015' datasets should also be archived."

*Response:*

*- All raw data that are not yet publicly available are uploaded to supplementary materials. Table II indicate which chronologies this is.*

*- The correct doi is inserted for Seftigen et al. 2015*

*- We note that there was a typo in the previous ms version. Table II should have 25 entries. The table and the text are now corrected.*